# Purified Acidic Sophorolipid Biosurfactants in Skincare Applications: An Assessment of Cytotoxic Effects in Comparison with Synthetic Surfactants Using a 3D In Vitro Human Skin Model

**Simms A. Adu** [1][ID]**, Matthew S. Twigg** [2]**, Patrick J. Naughton** [1][ID]**, Roger Marchant** [2] **and Ibrahim M. Banat** [2,*][ID]

1 The Nutrition Innovation Centre for Food and Health (NICHE), School of Biomedical Sciences, Faculty of Life and Health Sciences, Ulster University, Coleraine BT52 1SA, UK; adu-s@ulster.ac.uk (S.A.A.); pj.naughton@ulster.ac.uk (P.J.N.)
2 Pharmaceutical Science Research Group, Biomedical Science Research Institute, Ulster University, Coleraine BT52 1SA, UK; m.twigg@ulster.ac.uk (M.S.T.); roger.marchant@ulster.ac.uk (R.M.)
* Correspondence: im.banat@ulster.ac.uk; Tel.: +44-28-7012-3062

**Abstract:** Acidic sophorolipids (Acidic SL), congeners of sophorolipid biosurfactants, offer a potential alternative to synthetic sodium lauryl ether sulphate (SLES) in skincare applications. However, major challenges associated with the laboratory-based investigations of the cytotoxic effects of Acidic SL have been the utilisation of impure and/or poorly characterised congeners as well as the use of monolayers of skin cells in in vitro assays. While the former limitation makes glycolipids less attractive for use in academic research and skincare applications, the latter does not provide an accurate representation of the in vivo human skin. The present study, therefore, for the first time, assessed the cytotoxic effects of 96% pure Acidic SL on a 3D in vitro skin model in comparison with SLES, with the aim of investigating a natural alternative to synthetic surfactants for potential use in skincare applications. The 3D in vitro skin model was colonised with *Staphylococcus epidermidis* for 12 h, and afterwards treated with either Acidic SL or SLES at 100 µg mL$^{-1}$ for a further 12 h. Subsequently, the cytotoxic effects of Acidic SL in comparison with SLES were assessed using a combination of microbiology, molecular biology techniques, immunoassays, and histological analyses. It was demonstrated that Acidic SL had no deleterious effects on the viability of *S. epidermidis*, tissue morphology, filaggrin expression, and the production of inflammatory cytokines in comparison to SLES. These findings, in conjunction with the possibility to produce Acidic SL from cheaper renewable natural resources, demonstrate that Acidic SL could offer a potential sustainable alternative to synthetic surfactants.

**Keywords:** sophorolipids; synthetic surfactants; cytotoxicity; skin irritation; ecotoxicity skincare; sustainability; 3D in vitro skin model

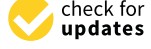



## 1. Introduction

The development of skincare routines with ingredients that have little or no deleterious effects on the healthy human skin and the skin microbiome, and/or with added functionalities such as skin surface moisturisation, microbiome restoration, and immunomodulation in diseased skin, is highly desirable in the formulation of skincare products [1–3]. However, surface active agents including sodium lauryl ether sulphate (SLES), which are a major component of many skincare formulations, are mostly synthesised from less sustainable and poorly degradable petrochemical resources [1,4,5]. In addition, these surfactants are reported to have the potential to cause skin irritation, allergic reaction, and skin microbiome dysbiosis, as well as contributing to ecotoxicity [2,4]. The adverse effects of synthetic surfactants on consumer skin health and the environment have necessitated investigations by the cosmeceutical and biotechnology industries into the production of biologically

derived surfactants (biosurfactants) from bacteria and yeasts via microbial fermentation as a substitute to synthetic surfactants in skincare formulations [1,4,6].

Microbial sophorolipids belonging to the glycolipid class of biosurfactants and produced by the yeast *Starmerella bombicola* have been reported in several studies as promising alternatives to synthetic surfactants [1,4,7–9]. The compelling advantages of sophorolipid biosurfactants over synthetic surfactants include enhanced foaming and solubilisation efficacy, low cytotoxicity, and added potential benefits such as wound healing and anticancer effects [7,10,11]. In particular, acidic sophorolipids (Acidic SL), which are a congener of sophorolipids, have been demonstrated to have little or no cytotoxic effects on human keratinocytes and skin dermal fibroblastic cells when assessed in vitro for potential use in skincare applications [7,8,12]. However, a major limitation to the skincare applications of sophorolipid biosurfactants has been the unwillingness to incorporate crude, impure, and/or poorly characterised sophorolipid congeners into skincare products [5,13]. Such a drawback stems from requirements by the EU Cosmetic Legislation (2013/674/EU) for manufacturers to specify the purity, molecular weight, and physio/bio-chemical properties of compounds used in formulations (extensively reviewed elsewhere by Adu and colleague) [5,13]. In addition, most studies on the cytotoxicity assessment of Acidic SL were conducted using 2D in vitro cultures; this allowed for the experimentation on a monolayer of cells only (e.g., keratinocytes and dermal fibroblasts) as opposed to the myriad tissues that make up the complex in vivo human skin [7,8,12,14–16]. Although 2D in vitro cell culture models are cheap, easy to use, and have contributed significantly to in vitro studies, they do not always provide an ideal representation of the in vivo human skin and, as such, may not translate into either pre-clinical in vivo animal models or first-in-human clinical trials [14,17,18].

The use of in vivo animal models as a further step on from in vitro 2D cell cultures is widely accepted as the de novo pathway for the generation of pre-clinical data; however, there are significant limitations. These include, but are not limited to, ethical considerations and the intrinsic differences in the anatomical structure, physiological functions, and host microbiota between human and animal models [19,20]. In agreement with the EU directive (Council Directive 76/768/EEC) on a proposed ban on the use of animal models for cosmeceutical ingredients testing, coupled with the limitations associated with the use of 2D in vitro cultures, the use of novel technologies to develop 3D in vitro skin models has become increasingly important [20–23]. The use of these 3D in vitro skin models allows for an appropriate representation of the complex anatomy and physiological functions of the in vivo human skin while providing an alternative to 2D in vitro skin cultures and the use of animal models in laboratory research [24,25].

At present, 3D skin models such as the Labskin™ full thickness 3D in vitro skin model (Labskin™), Phenion™ FT LongLife skin model, and Episkin™ are commercially available for testing the safety and efficacy of drugs and cosmeceutical ingredients [26]. Established among these models is Labskin™ [27,28]. Labskin™ is a full thickness human skin equivalent, specifically developed for studying interactions between the human skin and the skin microbiome [29]. The robust structured architecture of Labskin™ in addition to its well differentiated epidermis, barrier functions, and dry acidic surface makes it an ideal model for high-throughput studies on the effects of cosmetic ingredients on the interaction between the human skin and skin bacteria in vitro; hence, the choice for the present study [24,28].

Three-dimensional in vitro skin models have been utilised in a number of in vitro studies [21,28,30]. Holland and colleagues studied the colonisation of Labskin™ with *Staphylococcus aureus* and *Staphylococcus epidermidis* using microarray analysis and bacterial enumeration [19]. There was a ten-fold reduction in colonisation by *S. epidermidis* compared to *S. aureus* [19]. The microarray analysis revealed a significant upregulation in gene expression of beta-defensin and tumour necrosis factors compared to the uncolonised and *S. aureus*-colonised tissues [19]. Lewis and colleagues investigated the processes of in vitro wound healing and skin barrier repair using histological and matrix-assisted laser

desorption/ionization-mass spectrometry imaging (MALDI-MSI) analysis, following the creation of artificial wounds in a 3D in vitro skin model [30]. This study allowed for the identification of lipids, such as glycosylceramides, directly involved in skin barrier repair and wound healing in vitro [30]. Nevertheless, to the authors' knowledge, there is currently no study on the assessment of the effects of purified and chemically characterised sophorolipid biosurfactant congeners on 3D in vitro skin models for potential skincare applications in synthetic surfactants.

The aim of this study was to expand our previous investigations of natural alternatives to synthetically derived surfactants such as SLES for use in skincare applications, from 2D in vitro cell culture models to a pre-clinical 3D in vitro human skin model. To achieve this, we initially colonised the 3D in vitro human skin model with a bacterium representative of the human skin micro-flora (*S. epidermidis*) and then treated with either SLES or naturally derived non-acetylated acidic sophorolipids. The effect on the skin model of colonisation followed by surfactant treatment was assessed by measuring the bacterial viability of the infective strain, immunological markers of inflammation associated with bacterial infection and surfactant treatments, and direct observation either by H & E staining or IHC.

## 2. Materials and Methods

### 2.1. Chemical Characterisation of Acidic SL

Pre-purified non-acetylated acidic sophorolipids purchased from Biosynth Carbosynth, Compton, UK, were chemically characterised using HPLC–MS/ESI [31]. Stock preparations of Acidic SL and SLES were prepared (1 mg mL$^{-1}$) in sterile phosphate-buffered saline (PBS) (Merck, Gillingham, UK) and stored at $-20$ °C until required for use.

### 2.2. 3D In Vitro Skin Model Preparation

The 3D in vitro skin models used in this study, Labskin$^{TM}$ (Innovenn Ltd., York, UK), was supplied after 14 days of seeding, and was delivered in a 12-well cell culture insert as 12 skin constructs, each having a diameter of 1.1 cm (Figure 1). The constructs were partially suspended in Labskin$^{TM}$ maintenance medium (Innovenn Ltd., York, UK) containing 1% (*w/v*) agarose gel for transportation (Figure 1).

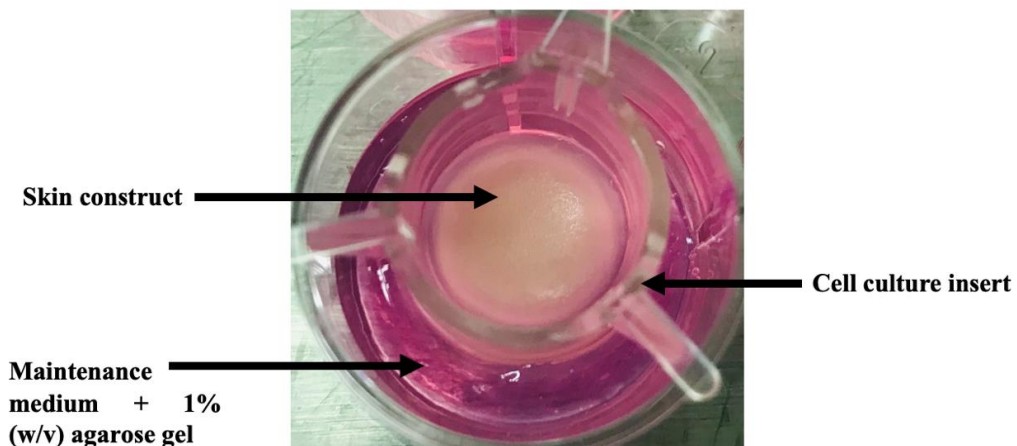

**Figure 1.** Uncolonised 1.1 cm diameter of the Labskin$^{TM}$ 3D in vitro skin model. The skin constructs were seeded in cell culture insert and transported in maintenance medium containing 1% (*w/v*) agarose gel.

The skin constructs were transferred into a ThinCert$^{TM}$ 12-well cell culture plate (Greiner Bio-One, Frickenhausen, Germany) containing 4.5 mL of pre-warmed maintenance medium and incubated for 48 h at 37 °C in 5% CO$_2$ (*v/v*) to equilibrate tissue metabolism. For the experiment, the skin constructs were transferred into a 12-well cell culture plate

(Sarstedt, Leicester, UK) containing 1 mL of fresh maintenance medium, followed by bacterial inoculation and surfactant treatments, respectively.

### 2.3. Colonisation of Skin Constructs with Staphylococcus Epidermidis

*Staphylococcus epidermidis* (DSM 28319), obtained from the German Collection of Microorganisms and Cell Cultures (Leibniz Institute DSMZ, Braunschweig, Germany) as a freeze-dried sample, was revived as per the supplier's instructions and stored in 25% glycerol (*v/v*) at $-80\,^\circ$C until required for use.

For the experiment, *S. epidermidis* was thawed on ice for 20 min (mins), cultured overnight in brain heart infusion broth (BHI) (Oxoid, 3124409, Hampshire, UK) at 37 $^\circ$C (primary seed culture), followed by the preparation of an exponential phase culture using the primary seed culture at a 1:10 dilution. Inocula prepared from the exponential phase culture were made in pre-warmed Dulbecco's phosphate-buffered saline (no calcium, no magnesium) (DPBS) (ThermoFisher Scientific, Loughborough, UK). Each construct was seeded at a volume of 10 μL (except for negative control constructs), with a final seeding density of $2.1 \times 10^5$ colony-forming unit per square centimetre (CFU cm$^{-2}$). Negative controls consisted of constructs with 10 μL of sterile DPBS only. Inocula were gently spread on the surface of constructs using an in-house-prepared sterile spreader (Glass Pasteur pipettes) (Scientific Laboratory Supplies Ltd., Lisburn, UK), and the tissues were incubated for 12 h at 37 $^\circ$C in 5% CO$_2$ (*v/v*) for skin surface colonisation.

### 2.4. Surfactant Treatments

The surfactants utilised in this study were SLES and pre-purified/chemically characterised Acidic SL. Post bacterial colonisation and tissue incubation for 12 h, three replicates each of the pre-colonised skin constructs were either untreated (Colonised) or treated with 100 μg mL$^{-1}$ of Acidic SL and SLES (Colonised + Acidic SL)/(Colonised + SLES) prepared in DPBS. Negative control was treated with 10 μL of sterile DPBS only (Uncolonised). The skin constructs were incubated for further 12 h at 37 $^\circ$C in 5% CO$_2$ (*v/v*), and then sampled for bacterial enumeration, tissue processing via formalin fixation and paraffin embedding, RNA extraction, and immunoassays.

### 2.5. Assessment of Bacterial Viability

The effects of Acidic SL and SLES on *S. epidermidis* pre-colonised skin constructs surface were assessed via bacteria enumeration using the drop plate method [32]. A total of 1 mL of pre-warmed DPBS was added to sampled 5 mm skin construct punch biopsies in a 1.5 mL Eppendorf tubes, vortex mixed for 2 min, and diluted to 1:10 in a fresh sterile DPBS. Subsequently, 10 μL was plated on BHI agar and incubated for 24 h at 37 $^\circ$C. The total viable bacteria (colonies) were enumerated and expressed as CFU per millilitre (CFU mL$^{-1}$). Further assessment for purity consisted of Gram's staining and phenotypic characterisation via plate streaking of isolated colonies on Mannitol Salt agar (MSA) (Oxoid, BCCB79883, Hampshire, UK) and Brilliance$^{TM}$ *E. coli*/Coliform agar (Oxoid, 1698837, Hampshire, UK). The bacterial enumeration experiments were performed independently three times, with each having five technical replicates.

### 2.6. Formalin Fixation, Tissue Processing, and Sectioning

Post 24 h of skin construct treatments (12 h of bacterial inoculation and 12 h of surfactant treatments), a 0.5 cm diameter of the skin construct from each treatment group was fixed in 10% formalin (Merck, Gillingham, UK) for 2 h, and further processed overnight using Leica TP1020 tissue processor (Leica, Nussloch, Germany) as per the manufacturer's instructions. The sampled tissues were embedded in Blue Ribbon$^{TM}$ paraffin wax (Leica, Nussloch, Germany), sectioned at 3 μm using Leica RM2145 rotary microtome (Leica, Nussloch, Germany), and transferred to Superfrost$^{TM}$ Plus adhesion glass microscope slides (Aquilant Scientific, Newtownards, UK). The tissues were partially dewaxed and left to firmly attach to the glass microscope slides by positioning the slides for drying on an

Epredia slide hotplate (ThermoFisher Scientific, Loughborough, UK) at 60 °C for 1 h prior to staining.

### 2.7. Haematoxylin and Eosin (H & E) Staining

Formalin-fixed, paraffin-embedded, and microtome-sectioned tissue samples were fully deparaffinised in HPLC grade xylene (Merck, Gillingham, UK) three times for a total of 9 min, followed by three washes in HPLC grade ethanol (Merck, Gillingham, UK) (9 min) and a rinse in tap running water for 2 min. The tissues were stained with Mayer's haematoxylin (Lillie's Modification) and Eosin Y solution (modified alcohol) (Abcam, Cambridge, UK) as per the manufacturer's instructions. The samples were mounted with DPX medium (Merck, Gillingham, UK) and viewed using a Digital Sight DS-U2 camera attached to an Eclipse E400 phase contrast microscope (Nikon Europe B. V., Amsterdam, The Netherlands) at 200× magnification. The thickness of the stratum corneum of the imaged tissues was analysed using the ImageJ Software version 1.53 for MacOS (ImageJ, National Institute of Health, Bethesda, MD, USA) [33–35].

### 2.8. Immunohistochemistry (IHC)

For IHC staining, the tissue samples were first processed, paraffin-embedded, sectioned at 3 μm, deparaffinised, and hydrated as detailed above. The antigen retrieval step consisted in heating the tissue samples to 95 °C for 20 min in an *in-house*-prepared citrate buffer (made with 10 mM anhydrous citric acid and 0.05% in 1000 mL distilled water and adjusted to pH 6.0). The tissues were washed ×3 with phosphate-buffered saline (PBS) (ThermoFisher Scientific, Loughborough, UK), and endogenous peroxidase activity was quenched by incubating the tissues with 100 μL of BLOXALL reagent (Vector Laboratories, Newark, CA, USA) per slide at room temperature (RT) for 10 min. Subsequently, the tissues were incubated for 20 min with 100 μL of normal horse serum, followed by 1 h incubation with 125 μL of prediluted recombinant anti-filaggrin primary antibody (1:500 dilution) (Abcam, ab221155, Cambridge, UK). Next, the tissues were incubated with 100 μL of prediluted biotinylated horse secondary antibody and incubated for 30 min at RT. Tissue staining was performed using Vectastain® Elite® ABC Universal Kit/ImmPACT® DAP peroxidase substrate (Vector Laboratories, Newark, CA, USA) as per the manufacturer's instructions. The tissues were counterstained with Mayer's haematoxylin (Lillie's Modification) (Abcam, Cambridge, UK) for 9 min, washed in running tap water for 2 min, differentiated in 0.25% acid alcohol for 2 s, and blued for 1 min. The tissue samples were dehydrated, cleared, mounted with DPX medium, and visualised as detailed in Section 2.7. The intensity of IHC staining was analysed using ImageJ Software, and reported as a percentage relative to the uncolonised tissue samples [33].

### 2.9. Enzyme-Linked Immunosorbent Assay (ELISA)

ELISA was utilised to investigate the effect of surfactants on the production of pro-inflammatory cytokines in the skin construct, using spent culture medium. The spent culture medium was collected into 1.5 mL Eppendorf tubes and stored at −80 °C until required for use. For the ELISA experiments, the spent culture medium was thawed to RT, centrifuged at 1000× $g$, and supernatants were analysed for Interlukin-8 (IL-8) and interlukin-6 (IL-6) cytokines using commercially available ELISA kits (R&D Systems, Inc., Minneapolis, MN, USA) as per the manufacturer's instructions. All ELISA experiments were performed independently three times.

### 2.10. RNA Extraction

Tissue punch biopsies (5 mm; approx. 120 mg), sampled post bacterial colonisation and surfactant treatment, were stored at −80 °C until required for use. For RNA extraction, briefly thawed (on ice) frozen sections of the skin constructs were first homogenised in 1 mL TRIzol™ Reagent (Invitrogen, Paisley, UK) on ice for 20 s using a VDI 12 stand model homogeniser (VWR Scientific, Lutterworth, UK) at setting 5. After homogenisation, total

RNA was extracted following the TRIzol™ Reagent manufacturer's protocol. The total RNA extracted was assessed for integrity using agarose gel electrophoresis and quantified using NanoDrop ND-1000 (ThermoFisher Scientific, Loughborough, UK).

### 2.11. cDNA Synthesis

The total RNA extracted from the skin constructs was reverse transcribed to generate cDNA samples using a G-STORM GS1 thermal cycler (Gene Technologies Ltd., Somerset, UK) [7]. Unless otherwise stated, all reagents utilised in the cDNA synthesis were sourced from ThermoFisher Scientific, Loughborough, UK. The reagents for the cDNA synthesis consisted of the following reaction mixtures with a total volume of 20 µL: 25 ng of Oligo (dT) 12–18 primer, 10 mM DTT, 0.5 mM dNTP, 50 ng of total RNA, 10 U of SuperScriptTM Reverse Transcriptase II (RT), and 12 µL molecular grade water (UltraPureTM distilled water). The cycling condition for the cDNA synthesis was set as follows: RNA denaturation step of 10 min at 70 °C, primer hybridisation step of 2 min at 42 °C, DNA polymerisation and subsequent cDNA synthesis step at 42 °C for 50 min, and, finally, an RT deactivation step at 70 °C for 15 min. Reverse transcriptase minus (NRT) negative control and No template negative control (NTC) were generated with the supplementation of the respective components.

To confirm that cDNA was successfully synthesised and free from genomic DNA contamination, and to validate the primer sets utilised in this study (Table S1), the synthesised cDNA was used as template DNA for endpoint PCR using a TC 500 Gradient Thermocycler (Techne, UK). The reaction mixture (total volume of 50 µL) for the endpoint PCR reaction consisted of 1 X PCR buffer, 1.5 mM MgCl2, 0.2 mM dNTP mix, 0.5 µM each of forward and reverse primer, 36 µL of molecular grade water (UltraPureTM distilled water), 2.5 U Taq polymerase (all reagents purchased from *ThermoFisher Scientific*, Loughborough, UK), and 1 µL of template DNA (synthesised cDNA). The cycling conditions for the endpoint PCR were set as follows: an initial denaturation step of 3 min at 94 °C, followed by 31 cycles of denaturation (95 °C for 30 s), a primer annealing step of 60 °C for 30 s, and an extension at 72 °C for 45 s. The final extension was held at 72 °C for 10 min.

### 2.12. Quantitative PCR (qPCR)

qPCR experiments were performed with synthesised cDNA from the skin constructs' total RNA, as detailed in a previous study [7], using the primer sets listed on Table S1. All qPCR experiments were performed independently three times. The data generated were analysed using the relative quantification method described by Maussion and colleagues [36]. The fold change, normalised to the reference gene (*β-actin*) relative to non-surfactant treated control, was calculated as $2^{-\Delta\Delta Cq}$ [36].

### 2.13. Statistical Analysis

A statistical analysis of all data was carried out using GraphPad Prism version 9.4.1 (458) for MacOS (GraphPad Software, San Diego, CA, USA). Data were analysed via a one-way analysis of variance (ANOVA), followed by Tukey's multiple comparison test; statistical significance was tested at $p \leq 0.05$ levels.

## 3. Results

### 3.1. Chemical Characterisation of Acidic SL

The sophorolipids reported in this study were obtained from a commercial source as pre-purified non-acetylated acidic sophorolipid congeners, and they were chemically characterised for congener profile and the relative percentage abundance of congeners present using HPLC–MS/ESI. The HPLC-MS/ESI analysis showed that the relative percentage abundance of congeners corresponding to non-acetylated acidic sophorolipid congeners were 96.41%, whereas the remaining 3.59% of the congeners reported corresponded to the mass of Lactonic SL congeners (Table 1). The predominant peak on the Acidic SL spectrum had ion $m/z$ of 621.3, corresponding to the molecular ion [M+H] + of the Acidic SL congener

Acidic, C18:1, which accounted for a relative abundance of 43.81% of the sample (Table 1, Figure 2).

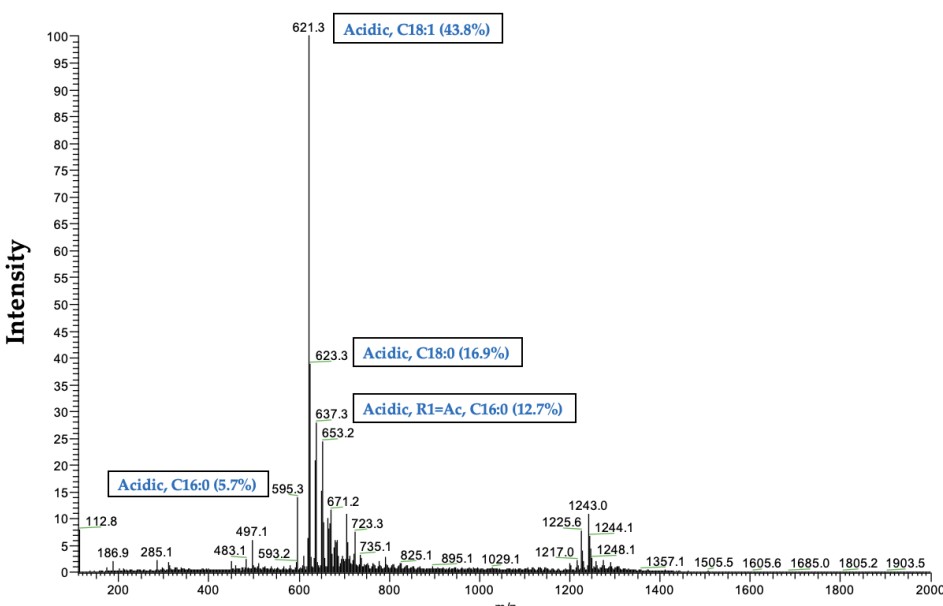

**Figure 2.** ESI-MS of identified peaks resulting from the HPLC separation of pre-purified Acidic SL.

**Table 1.** Identification and relative quantification of individual sophorolipid congeners as detected by the HPLC-MS/ESI analysis of the Acidic SL preparation used for skin construct treatment.

| Probable Structure | Observed *m/z* | Peak Area | % Relative Abundance |
|---|---|---|---|
| **% Abundance of Acidic SL congeners—96.41** | | | |
| Acidic, C16:0 | 595.3 | 158,650,232.27 | 5.74 |
| Acidic, C18:2 | 619.2 | 70,540,287.99 | 2.55 |
| Acidic, C18:1 | 621.3 | 1,210,090,321.46 | 43.81 |
| Acidic, C18:0 | 623.3 | 465,744,019.84 | 16.86 |
| Acidic, R1 = Ac, C16:0 | 637.3 | 350,794,715.78 | 12.70 |
| Acidic, 1Ac, C18:0 | 665.5 | 37,939,611.09 | 1.37 |
| Acidic, R1 = Ac, C18:1 or Acidic, R2 = Ac, C18:1 | 663.3 | 102,670,879.63 | 3.72 |
| Acidic, R1 + R2 = Ac, C16:0 | 679.2 | 29,300,132.66 | 1.06 |
| Acidic, R1 + R2 = Ac, C18:2 | 703.1 | 17,417,772.37 | 0.63 |
| Acidic, R1 + R2 = Ac, C18:1 | 705.3 | 126,826,959.83 | 4.59 |
| Acidic, R1 + R2 = Ac, C18:0 | 707.3 | 57,366,159.94 | 2.08 |
| Acidic, R1 = Ac, C22:0 | 721.2 | 15,293,285.62 | 0.55 |
| Acidic, R1 + R2 = Ac, C24:0 | 791.3 | 20,615,084.25 | 0.75 |
| **% Abundance of Lactonic SL congeners—3.59** | | | |
| Lactonic, R1 + R2 = Ac, C18:2 or Lactonic, R1 + R2 = Ac, C18:2 | 685.1 | 69,269,763.09 | 2.51 |
| Lactonic, R1 + R2-Ac, C18:1 or Lactonic, R1 + R2-Ac, C18:1 or Lactonic, R1 + R2 = Ac, C18:1 | 687.1 | 29,782,951.00 | 1.08 |

*3.2. Effects of Acidic SL and SLES on the Viability of S. epidermidis Colonised on Skin Constructs*

The effects of surfactants on the viability of *S. epidermidis* were assessed by comparing the number of colonies of *S. epidermidis* on the bacterial-colonised and non-surfactant-treated skin constructs with the constructs colonised with *S. epidermidis* and treated with surfactants. As expected, no colonies of *S. epidermidis* were detected on BHI agar for the skin constructs not colonised with *S. epidermidis* (Figure 3). There was no significant difference in the number of *S. epidermidis* colonies between the non-surfactant-treated constructs and the constructs treated with 100 µg mL$^{-1}$ of either Acidic SL or SLES (Figure 3). Moreover, there was no significant difference between the number of *S. epidermidis* colonies in Acidic SL- and SLES-treated skin constructs (Figure 3). Potential damages to the surface of the skin constructs post bacterial colonisation and surfactant treatment were examined macroscopically and microscopically, and the 3D in vitro skin model was found to be intact.

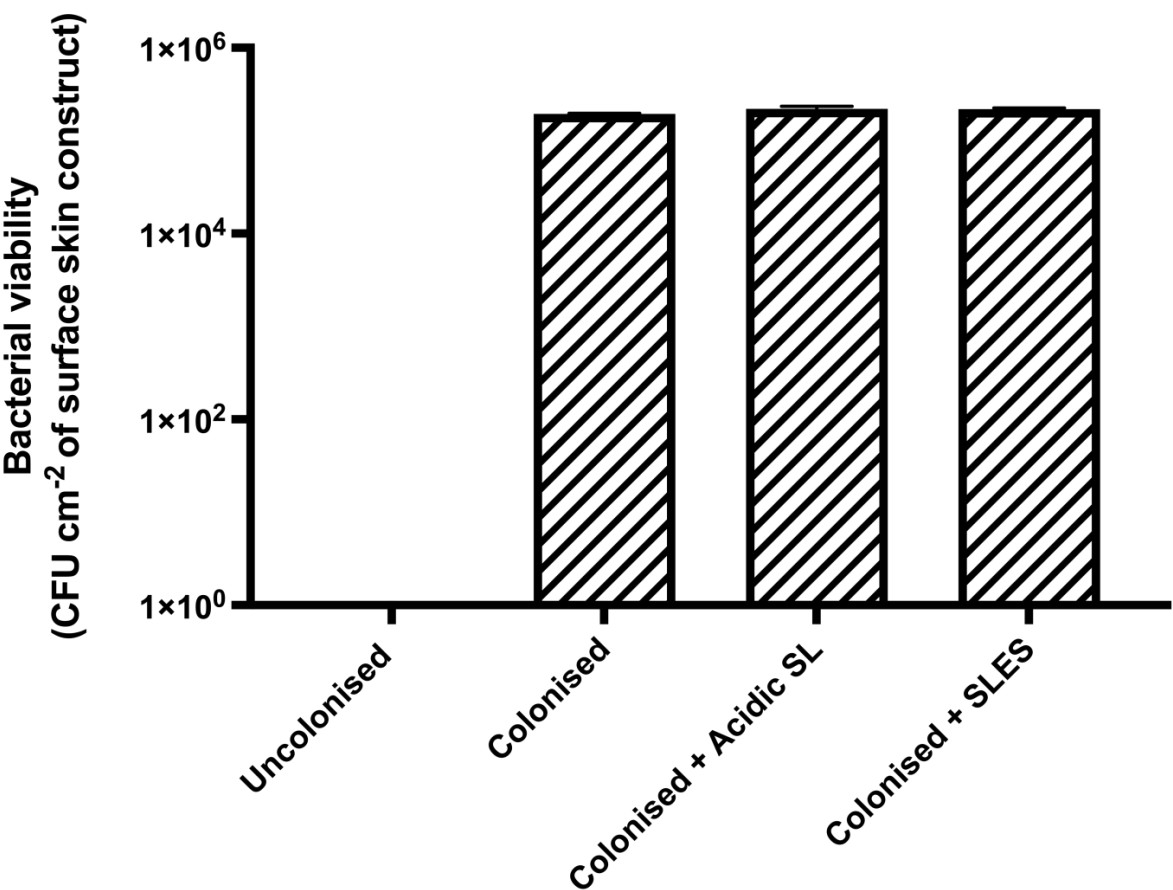

**Figure 3.** Analysis of *S. epidermidis* viability via the drop plate method. Skin constructs were either uncolonised, colonised with *S. epidermidis* and not treated with surfactants (Colonised), or colonised and further treated with 100 µg mL$^{-1}$ of Acidic SL (Colonised + Acidic SL) or SLES (Colonised + SLES). Data are the mean results of triplicate experiments. Error bars represent standard error from the mean. Statistical significance was determined using a one-way ANOVA followed by Tukey's multiple comparison test, and statistical significance was tested at $p \leq 0.05$ levels.

To determine whether *S. epidermidis* on BHI agar was free of microbial contaminants, the colonies of *S. epidermidis* were assessed for purity using Gram's staining and phenotypic characterisation via plate streaking on selective/differential media. Gram's staining revealed purple cocci grape-like clusters of approximately 1 µm diameter per cell size (Figure S1). The colonies of *S. epidermidis* appeared pale pink on MSA; however, on Brilliance$^{TM}$ *E. coli*/Coliform agar no bacterial growth was observed (Figure S1).

### 3.3. Effects of Acidic SL and SLES on the Morphology of Pre-Colonised Skin Constructs

Histological analysis revealed a normal skin structure with intact epidermis and evenly distributed fibroblasts in the dermal layer of both the untreated tissues and the tissues treated with 100 µg mL$^{-1}$ of either Acidic SL or SLES (Figure 4). Ulcerations, tissue erosion, and necrosis were absent in both the untreated and surfactant-treated tissues (Figure 4). Epidermal layers including the stratum corneum (SC), stratum granulosum (SG), stratum spinosum (SS), and stratum basale (SB) were identifiable in all treatment groups (Figure 4). Comparing SLES-treated tissues with those of Acidic SL, there were no observable differences in tissue morphology (Figure 4).

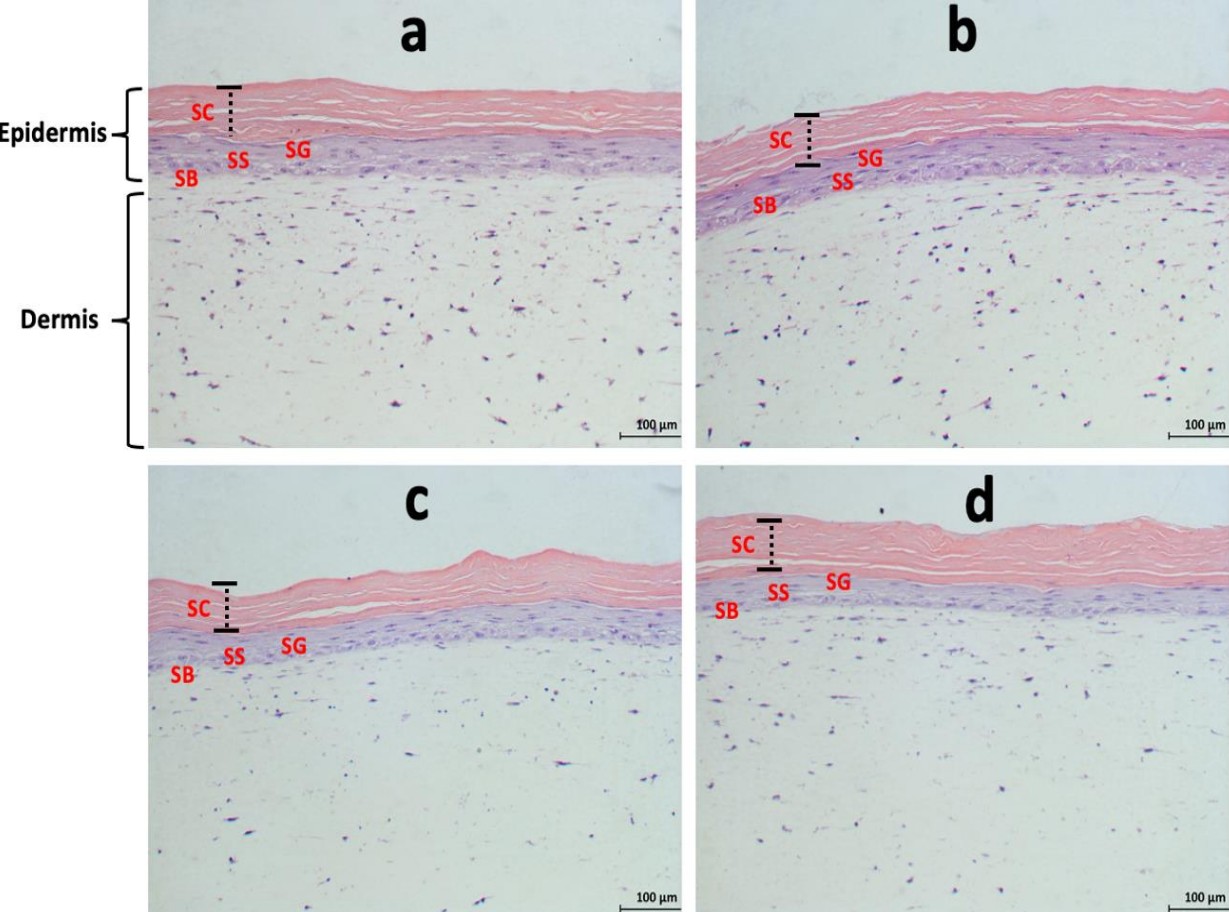

**Figure 4.** H & E staining of (**a**) tissue samples neither colonised nor treated with surfactants, (**b**) tissues colonised with *S. epidermidis* and not treated with surfactants, and (**c**) tissues colonised with *S. epidermidis* and further treated with 100 µg mL$^{-1}$ of Acidic SL or (**d**) SLES. Post H & E staining, the tissues were imaged using a Digital Sight DS-U2 camera attached to an Eclipse E400 phase contrast microscope at 200× magnification. Epidermal layers such as stratum corneum (SC), stratum granulosum (SG), stratum spinosum (SS), and stratum basale (SB) are identifiable in all treatment groups. Also, the dermal component marked by a spindled-cellular arrangement with oval nuclei (purple dots) remained intact. Three images per well were randomly selected for presentation; the scale bar was set at 100 µm.

The measured SC thickness of the uncolonised tissues was 66.97 µm (±2.32 µm), *S. epidermidis*-colonised and non-surfactant-treated tissue was measured at 67.21 µm (±0.59 µm), colonised and Acidic SL-treated tissue was 67.88 µm (±0.60 µm), and that of colonised and SLES-treated tissue was 67.00 µm (±1.53 µm). There was no significant

difference in SC thickness between surfactant-treated and untreated tissues ($p > 0.9999$), or between the Acidic SL- and SLES-treated tissues ($p = 0.9924$).

### 3.4. Effects of Acidic SL and SLES on Filaggrin Expression in the Epidermis of Skin Constructs

As previously demonstrated for the H& E staining, the tissue samples under all treatment conditions (uncolonised, colonised, colonised and Acidic SL-treated, or colonised and SLES-treated samples) maintained intact morphology, with each showing distinct skin epidermal and dermal layers (Figure 5). The topmost layer of sectioned skin construct epidermis (SC) under all treatment conditions was stained brown, indicative of FLG immunostaining (Figure 5).

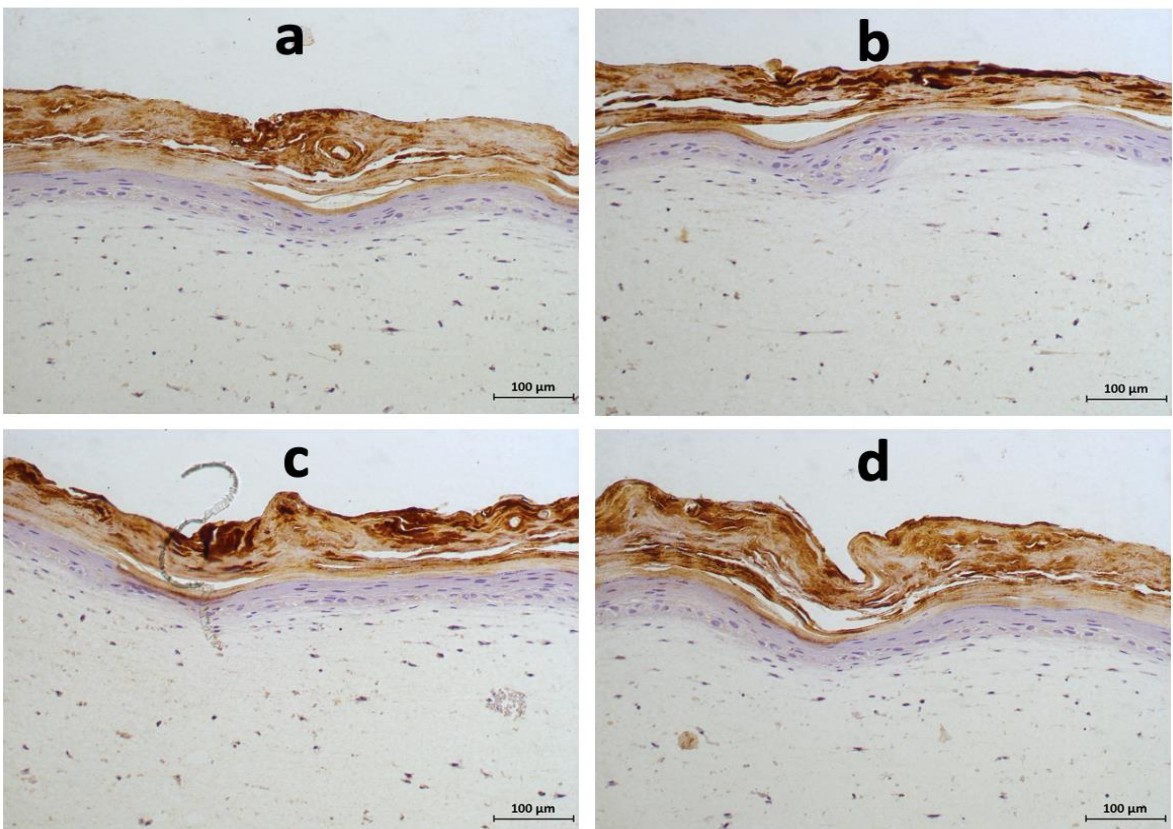

**Figure 5.** FLG stain intensity in the SC of skin construct epidermis. IHC staining of FLG in (**a**) tissues neither colonised nor treated with surfactants, (**b**) tissues colonised with *S. epidermidis* and not treated with surfactants, and (**c**) tissues colonised with *S. epidermidis* and treated with 100 μg mL$^{-1}$ of Acidic SL or (**d**) SLES. The tissues were imaged at 200× magnification. Three images per well were randomly selected for presentation; the scale bar was set at 100 μm.

The analysis of FLG immunostaining revealed that there was no significant difference in FLG stain intensity between the non-surfactant-treated tissues and tissues pre-colonised with *S. epidermidis* and treated with Acidic SL or SLES (Figure 6). Furthermore, there was no significant difference in stain intensity between the Acidic SL- and SLES-treated tissues (Figure 6).

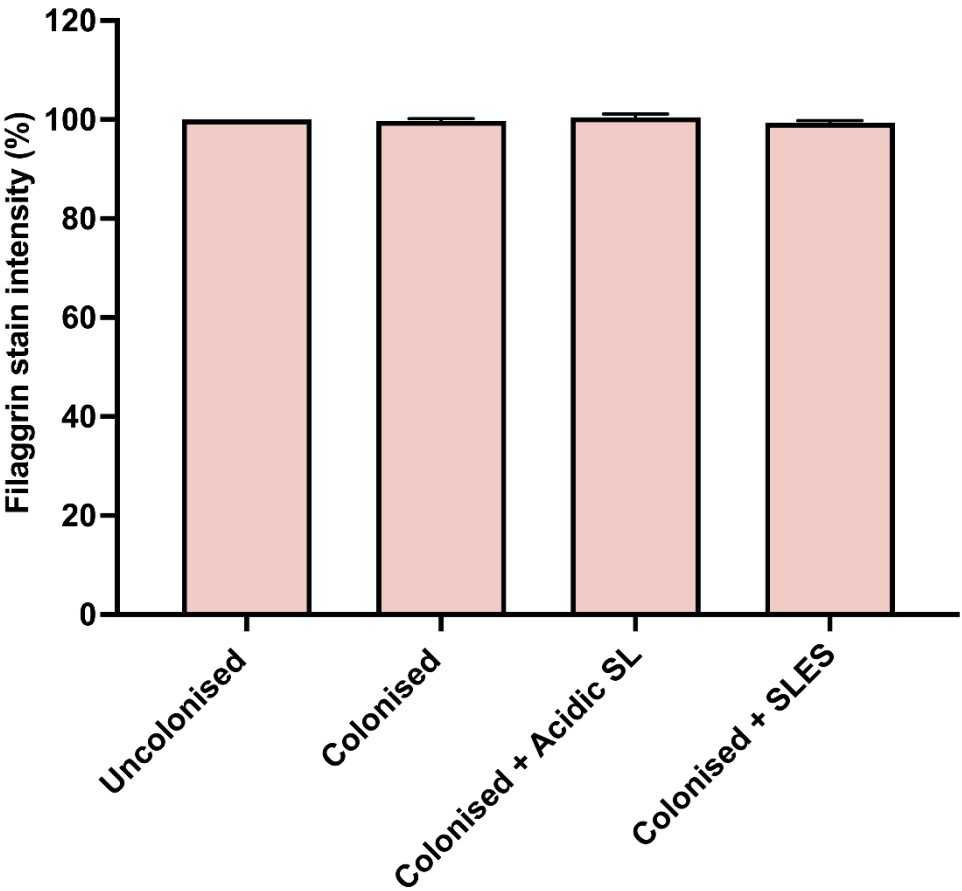

**Figure 6.** Analysis of FLG stain intensity in the SC of skin construct epidermis. The stain intensity of FLG in the skin epidermis was analysed using ImageJ software. The FLG stain intensity was expressed as a percentage relative to the uncolonised tissue samples. Data are the mean results of triplicate experiments; error bars represent standard error from the mean. Statistical significance was determined using a one-way ANOVA followed by Tukey's multiple comparison test, and statistical significance was tested at $p \leq 0.05$ levels.

*3.5. Effects of Acidic SL and SLES on Pro-Inflammatory Cytokine Production in Skin Constructs*

The effects of Acidic SL and SLES on the production of pro-inflammatory cytokines in the skin constructs pre-colonised with *S. epidermidis* were investigated via ELISA using spent culture medium. There was no significant difference in IL-8 and IL-6 levels between uncolonised and non-surfactant-treated samples, or between the *S. epidermidis*-colonised and non-surfactant-treated samples (Figure 7a,b). Compared to the control samples, a trend towards decreased IL-8 and IL-6 levels was observed in Acidic SL-treated samples; however, this was not statistically significant (Figure 7a,b). There was no significant difference in IL-8 and IL-6 production between the experimental controls and SLES-treated samples (Figure 7a,b). By directly comparing the levels of IL-8 and IL-6 produced in Acidic SL-treated samples with SLES, no significant differences were observed (Figure 7a,b).

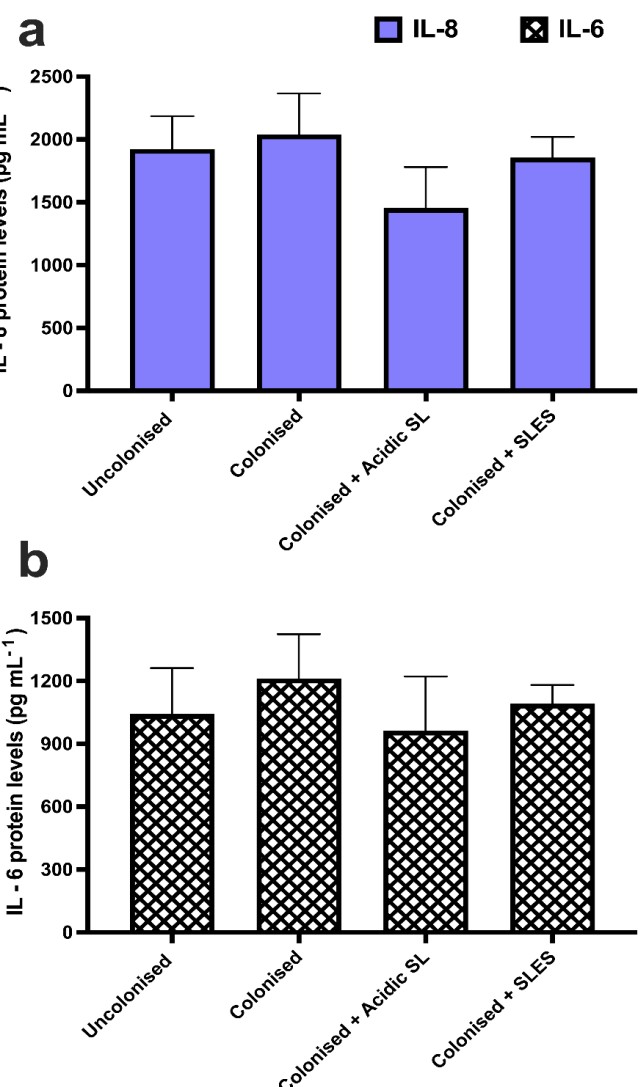

**Figure 7.** ELISA analysis of (**a**) IL-8 and (**b**) IL-6 production in the skin constructs. The levels of cytokine production were determined via ELISA using spent culture medium of the skin constructs neither colonised nor treated with surfactants (Uncolonised), constructs colonised with S. epidermidis and not treated with surfactants (Colonised), and skin constructs colonised with S. epidermidis and treated with 100 μg mL$^{-1}$ of Acidic SL (Colonised + Acidic SL) or SLES (Colonised + SLES). Data are the mean results of triplicate experiments; error bars represent standard error from the mean. Statistical significance was determined using a one-way ANOVA followed by Tukey's multiple comparison test, and statistical significance was tested at $p \leq 0.05$ levels.

### 3.6. Effects of Acidic SL and SLES on Pro-Inflammatory Cytokine Gene Expression in Labskin$^{TM}$

NanoDrop spectrophotometric analysis revealed that none of the samples extracted yielded less than 500 ng μL$^{-1}$ total RNA and OD 260/280 ratio less than 1.93. The gel electrophoresis analysis showed three non-smearing bands, i.e., 28S rRNA with an amplicon size of 2000 bp, 18S rRNA at 1000 bp, and 5.8S/5 rRNA at 100 bp, indicative of non-degraded RNA (Figure S2). Similarly, cDNA synthesised using total RNA was validated using endpoint PCR and, subsequently, the assessment of primer amplification efficiencies (Figures S3 and S4). The amplification efficiency of *β-actin* utilised as reference gene was 86.4%, whereas those of *CXCL8* and *IL-6* (utilised as genes of interest) were 92.3% and 91%, respectively (Figure S4).

The effects of Acidic SL and SLES on the expression of pro-inflammatory cytokine genes were assessed using RT-qPCR. Compared to uncolonised and untreated skin con-

structs, increased levels of *CXCL8* gene expression, as measured by fold change in expression relative to a reference gene (*β-actin*), was observed in *S. epidermidis*-colonised untreated and Acidic SL-treated tissues. In both cases, the difference in gene expression was not statistically significant (Figure 8a). *S. epidermidis*-colonised and SLES-treated constructs showed a reduced level of *CXCL8* gene expression compared to constructs colonised and treated with Acidic SL; again, the difference in expression was not significant (Figure 8a). A similar non-significant trend in the expression of *IL6* (also measure by fold change in expression relative to the reference gene *β-actin*) was also observed between each treatment group (Figure 8b).

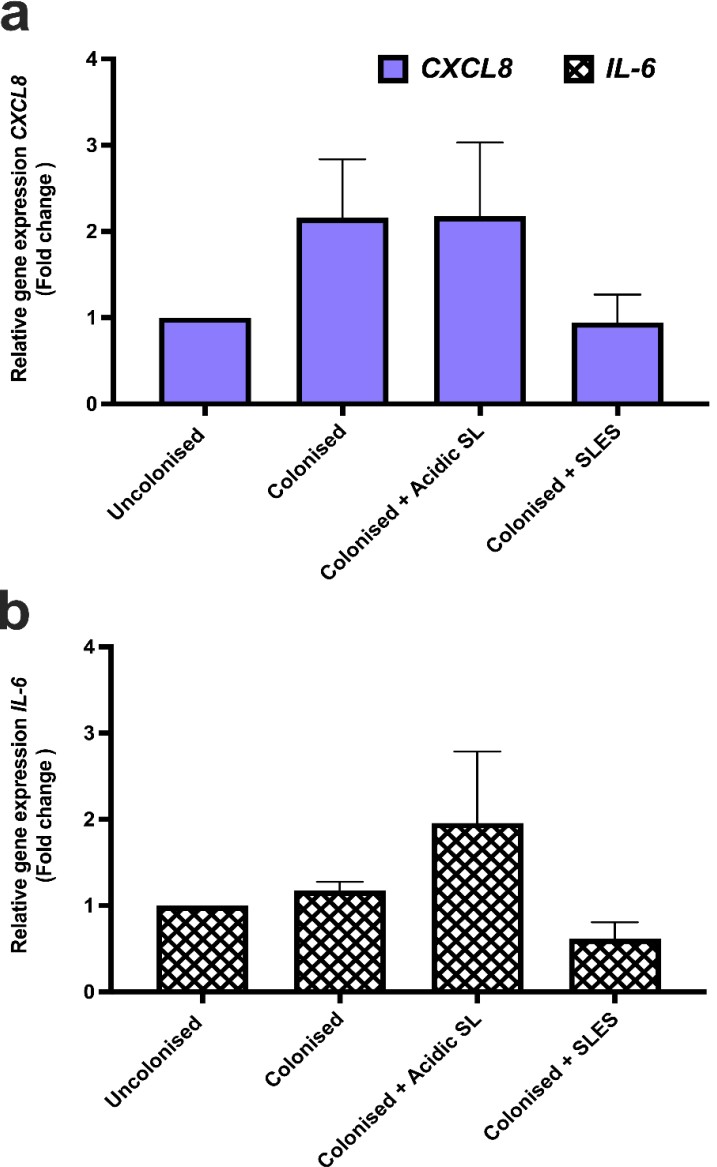

**Figure 8.** qPCR analysis of (**a**) *CXCL8* and (**b**) and *IL-6* expression in the skin construct. Tissues were either uncolonised, colonised with *S. epidermidis* and not treated with surfactants (Colonised), or colonised and further treated with 100 µg mL$^{-1}$ of Acidic SL (Colonised + Acidic SL) or SLES (Colonised + SLES). Data are the mean results of triplicate experiments; error bars represent standard error from the mean. Statistical significance was determined using a one-way ANOVA followed by Tukey's multiple comparison test, and statistical significance was tested at $p \leq 0.05$ levels.

## 4. Discussion

The use of robust novel experimental and instrumental techniques which allow for accurate mimicking of the in vivo human system is crucial for the growing skincare industry [24]. Three-dimensional in vitro skin models are currently being used as a replacement for 2D in vitro models, ex vivo, and animal models in skin research, given that 3D in vitro skin models provide a more accurate representation of the anatomy and physiological functions of the in vivo human skin [27,28]. The present study assessed the cytotoxic effects of 96% pure preparations of Acidic SL on Labskin[TM] full thickness 3D in vitro skin model pre-colonised with *S. epidermidis* in comparison with SLES. *S. epidermidis* was utilised in the present study since most skin commensals are Gram-positive bacterial species, and among these, *S. epidermidis* is one of the most prevalent isolates at the dry niches of the human skin, with the potential to produce antimicrobials (e.g., antibiotics) to inhibit the colonisation of pathogenic skin bacteria [37,38]. Also, the choice of the surfactants and treatment concentration utilised in this study was based on the differential effects of Acidic SL and SLES on the viability and morphology of the 2D in vitro skin cell cultures reported in our previous studies in accordance with the Organisation for Economic Cooperation and Development's (OECD) guidelines on toxicological studies [7,22,23,31,39]. In these studies, the Acidic SL were less cytotoxic compared with SLES, in that whereas up to 500 µg mL$^{-1}$ Acidic SL demonstrated no significant cytotoxic effects on cell viability and morphology, the viability of cells was significantly reduced when treated with SLES concentrations above 60 µg mL$^{-1}$ [7,31]. Hence, the use of these two surfactants in the current study allowed for a further investigation of a sustainable and natural-skin-compatible alternative to SLES for skincare applications using a 3D in vitro skin model.

Bacterial enumeration, histological analysis, and the use of a combination of immunoassays and molecular biology techniques in the present study revealed that at the concentration tested (100 µg mL$^{-1}$), the purified Acidic SL had no deleterious effects on the viability of *S. epidermidis*, tissue morphology, FLG expression, and the production of inflammatory cytokines comparable to SLES. To the best of the authors' knowledge, this is the first time 3D in vitro skin models have been utilised to assess the cytotoxic effects of purified and chemically characterised sophorolipid congeners for potential skincare applications.

The HPLC-MS/ESI analysis of the sophorolipid preparation utilised in this study revealed that Acidic SL congeners comprised 96% of the sample. It is important to understand that Acidic and Lactonic SL congeners can possess completely different bioactivities and that several previous in vitro studies on SL bioactivities utilised preparations possessing a less defined mixture of both Acidic and Lactonic congeners [8,40,41]. Consequently, studies that utilised mixed/impure congeners of sophorolipids resulted in significant interstudy variations as well as contaminant-associated toxicity, which in effect renders sophorolipids less attractive for potential use in skincare and pharmaceutical applications [42,43]. Consistent with the congener profile of the Acidic SL reported in this study, Lydon and colleagues utilised HPLC-MS analysis to characterise Acidic SL for use in the assessment of wound healing and adjuvant antimicrobial properties [12]. The congeners of sophorolipids present in their extracts corresponding to Acidic SL congeners were above 95% [12]. In another study, Callaghan and colleagues utilised HPLC coupled with evaporative light scattering detector (HPLC-ELSD) and HPLC-MS analyses to characterise Lactonic SL [42]. In that study, Lactonic SL congeners comprised 97% of the sample used for testing bioactivity against cancer cell lines [42]. This clearly indicates the importance of utilizing preparations with a high percentage relative abundance of each chemical sub-species of sophorolipid, (Acidic SL or Lactonic SL), to generate data on discernible bioactive effects on both the prokaryotic and eukaryotic cells that are attributable to the respective SL sub-species used [12,42].

The human skin surface in in vivo systems is not sterile, but rather colonised with commensal and mutualistic microorganisms that improve skin health. Hence, the 3D in vitro skin model utilised in the current study was first colonised with a bacterial species

dominant within the human skin commensal population, *S. epidermidis*, and subsequently treated with either Acidic SL and SLES to assess the effects of these surfactants on bacterial viability and interaction with the skin constructs. Neither Acidic SL nor SLES had any significant inhibitory effects on the viability of *S. epidermidis*. Previous studies on the in vitro interaction between sophorolipids and bacterial cells focused on antimicrobial effects of sophorolipids rather than their compatibility effects on the representatives of the normal human skin microbiota [40,44]. Sophorolipid mixtures with varying sugar head groups were demonstrated to have antimicrobial effects, mainly on Gram-positive bacterial cells such as *Streptococcus agalactiae* and *Bacillus subtilis* at concentrations as low as 0.024 mg mL$^{-1}$; this is in contrast to the results reported in the current study in that no inhibitory effects were observed here [44]. It is, however, worthy of note that the antimicrobial effects of sophorolipids are mainly dependent on their purity, molecular structure, physiochemical properties, properties of bacterial cell membrane, and the mechanism of the sophorolipids' interaction with the bacterial cell membrane [11,40,45]. Therefore, although all bacterial isolates reported here are Gram-positive, differences in the purity and physiochemical properties of the sophorolipids utilised in the present study compared to the sophorolipid mixtures reported by Shah and colleagues may have contributed to the contrasting dose–response [44]. This notwithstanding, skincare ingredients with non-deleterious effects on the healthy human skin microbiome are highly sought after for various skincare applications [10,11]. Therefore, the non-inhibitory effects of the purified Acidic SL on skin bacteria as demonstrated in the present study, coupled with added functionalities such as foaming and solubilisation properties, makes them ideal candidates for incorporation into skincare products. While a single bacterium does not represent the entirety of the complex human skin microbiome, the preliminary findings that Acidic SL has non-deleterious effects on *S. epidermidis* pre-colonised on 3D in vitro skin model is an important step towards understanding the effects of sophorolipids on the in vivo human skin microenvironment.

In terms of the effects of Acidic SL on the morphology of the skin model comparative to SLES, no deleterious effects were observed when the tissues were treated with surfactants, stained with H & E, and visualised using phase contrast microscope. Measurement of epidermal thickness is a well-known method for determining the severity of skin inflammation in skin diseases such as atopic dermatitis [34]. Here, neither Acidic SL nor SLES had any effects on the thickness of the measured stratum corneum of the skin constructs. The cytotoxic effects of sophorolipids are hypothesised to be dependent on their level of acetylation [46]. Hence, sophorolipids with high levels of acetylation are proposed to have higher cytotoxic effects, as the acetyl groups on their hydrophilic moiety may have high penetrative effects on cells/tissues [45,46]. In the present study, the Acidic SL utilised were non-acetylated, and this could account for the non-deleterious effects on the 3D in vitro skin model. Another important consideration for the non-deleterious effects of the Acidic SL reported in the current study could be the characteristics of the tissue model system utilised. It has been previously demonstrated that whereas 60 µg mL$^{-1}$ of SLES significantly reduces the viability of a monolayer of spontaneously transformed human keratinocytes (HaCaT cells), Acidic SL at treatment concentrations up to 500 µg mL$^{-1}$ have no significant effects on the viability and morphology of HaCaT cells [7,12]. Although the dose effects of Acidic SL utilised in the above studies agree with the present study, there is a contrast in the dose–response of SLES in that no deleterious effects were observed on the skin model with the treatment at 100 µg mL$^{-1}$ as opposed to the 60 µg mL$^{-1}$ treatment concentration, which significantly reduced the viability of HaCaT cells [7]. The differences in structural architecture and physiological properties between the monolayer of cells utilised in the studies above and the 3D in vitro skin model reported in the current study may have accounted for this discrepancy [7,12,14]. In particular, monolayers of HaCaT cells lack fibrin matrixes, which influence the robust structural architecture of the 3D in vitro skin models such as the one utilised in the present study [28]. This brings to bear potential future study limitations in transitioning from the use of 2D to 3D in vitro

skin models considering the difference in structural architecture of these two separate models and the subsequent differences in response to therapeutic agents. Nevertheless, further studies using a broad range of treatment concentrations and employing additional analytical techniques will be required to observe discernible differences in dose effects between sophorolipid biosurfactants and synthetic surfactants when utilising 3D in vitro skin models.

FLG is a structural protein in the SC of the skin epidermis [47,48]. FLG is fundamental in controlling the shape of keratinocytes, the maintenance of the texture of the epidermis, the promotion of the aggregation of intermediate filaments and the subsequent formation of mechanically resilient corneocyte intracellular matrix, and the control of SC permeability and hydration [49–51]. Consequently, a deficiency in filaggrin expression in the stratum corneum of the skin epidermis could lead to impaired skin barrier functions and an exacerbation of non-lesion atopic dermatitis [50,52,53]. Tö Rmä and colleagues demonstrated that the exposure of eleven healthy volunteers to 50 μL of 1% ($v/v$) SLES for six h significantly reduced the expression of pro-filaggrin in human keratinocytes [54]. In the present study, a non-significant effect of both Acidic SL and SLES on FLG expression was reported. The difference in dose–response in the present study compared to the report by Tö Rmä and colleagues could be explained by the difference in the concentration of SLES utilised (0.01% versus 1% ($v/v$)) [54]. This notwithstanding, the results reported in the current study suggest that at the tested concentration, Acidic SL may have non-deleterious effects on the architectural structure of the human skin epidermis and on the barrier functions of the human skin comparable to SLES.

Similar to the effects of Acidic SL compared with SLES on tissue morphology and FLG expression in the 3D in vitro skin model, the use of ELISA and qPCR revealed that neither surfactant utilised in the present study had any significant effects on IL-8/*CXCL8* and IL-6/*IL-6* stimulation compared to control samples. It was interesting to observe a trend drop in IL-8 and IL-6 levels in Acidic SL-treated tissues, indicative of potential immunomodulatory effects; however, this was not statistically significant. Such a trend drop was not observed in SLES-treated tissues or in similarly purified Acidic SL employed in other studies when a monolayer of human keratinocytes was utilised [7]. Further studies focusing on the potential immunomodulatory effects of Acidic SL and associated mechanisms of action in 3D in vitro skin models will be an important future step.

It should be noted that sophorolipids could only be used as a substitute to synthetic surfactants if they are able to demonstrate equal or better performance in skincare formulation at a reasonable market price [55,56]. Therefore, having demonstrated that the Acidic SL utilised in the present study have comparable effects with SLES on the 3D in vitro skin model, in addition to being produced from sustainable natural resources (e.g., industrial waste materials such as vegetal cooking oil waste), and having enhanced solubilisation and foaming functions, Acidic SL could offer a potential natural/sustainable alternative to SLES in skincare applications [5,56].

## 5. Conclusions

In comparison to the synthetic surfactant SLES, naturally derived Acidic SL demonstrated no deleterious effects on the viability of pre-colonised *S. epidermidis*, tissue morphology, FLG expression, and the production of pro-inflammatory cytokines in a pre-clinical in vitro 3D skin model. This study demonstrates an early proof of concept on the potential use of purified Acidic SL congeners in skincare applications at concentrations that would not be detrimental to the healthy human skin and bacteria, hence paving the way for further in vitro and in vivo studies on healthy human skin and the skin microbiota using purified sophorolipid biosurfactant congeners. To the best of the authors' knowledge, this is the first time the cytotoxic effects of purified Acidic SL have been assessed in comparison with synthetic surfactants using a full thickness 3D in vitro skin model. There are, however, specific limitations to the present study, which are worth pointing out. Firstly, despite the simplicity, reliability, and reproducibility of the 3D in vitro model utilised, it

currently lacks several important in vivo skin components such as blood vessels, hair, hair follicles, sweat, and sebaceous glands. The incorporation of these complex in vivo skin components into this 3D in vitro model has been deemed unfit for future research due to high production costs and limited technical expertise. Secondly, considering the high cost of developing/acquiring the 3D in vitro skin model, the present study was limited to the use of a single bacterium (*S. epidermis*) and one surfactant treatment concentration condition (100 µg mL$^{-1}$). The high monetary cost of the model also limited the number of experimental repeats that could be carried out in the current study and may explain why much of the data was only trending and not significantly significant. The financial cost of generating robust datasets using this model in comparison to using an in vivo animal model must be of significant consideration when choosing a suitable pre-clinical model for the testing for naturally derived compounds such as Acidic SL. Future studies should focus on identifying a broad range of treatment concentrations to conduct dose–response experiments on the bioactivities of sophorolipid biosurfactants when using a 3D in vitro skin model.

**Supplementary Materials:** The following supporting information can be downloaded at: https://www.mdpi.com/article/10.3390/fermentation9110985/s1, Figure S1: Phenotypic characterisation of sampled *S. epidermidis*; Figure S2: Gel electrophoretic analysis of total RNA extracted from Labskin$^{TM}$; Figure S3: Gel electrophoretic analysis of cDNA synthesised from total RNA extracted from Labskin$^{TM}$; Figure S4: Standard curves and primer efficiencies of primer sets utilised in qPCR analysis; Table S1: Sequence of primer sets and product size of pro-inflammatory markers utilised in endpoint PCR and qPCR analyses.

**Author Contributions:** S.A.A. conducted experiments and wrote the manuscript. M.S.T. analysed mass spec data and edited the manuscript. P.J.N., R.M. and I.M.B. conceived, designed, and supervised the research. All authors have read and agreed to the published version of the manuscript.

**Funding:** This work was supported by a Vice Chancellors Research Scholarship awarded to S.A.A. by Ulster University through NICHE. Additional support was obtained from Invest Northern Ireland, U.K., proof of concept grant number 826.

**Informed Consent Statement:** Not applicable.

**Data Availability Statement:** The datasets generated during and/or analysed during the current study are available from the corresponding authors on reasonable request.

**Acknowledgments:** The authors would like to thank the Mass Spectroscopy Unit at Ulster University for their assistance with the chemical analyses of Acidic SL. The authors would also like to thank Labskin$^{TM}$ UK Ltd. for offering SAA training in the appropriate handling of a full thickness 3D in vitro skin model and subsequent histological sectioning and staining.

**Conflicts of Interest:** The authors declare no conflict of interest.

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
