# Peer review of "Purified Acidic Sophorolipid Biosurfactants in Skincare Applications: An Assessment of Cytotoxic Effects in Comparison with Synthetic Surfactants Using a 3D In Vitro Human Skin Model"

_fermentation, doi:10.3390/fermentation9110985_

Round 1

Reviewer 1 Report

Comments and Suggestions for Authors

The authors investigated the cytotoxic effects of Acidic SL on a 3D in-vitro skin model in comparison with SLES for investigating a natural alternative to synthetic surfactants for potential use in skincare applications. The idea of testing new substances on in vitro models without the use of animals is in line with EU recommendations and ethical expectations. However, the use of such a model (in this case Labskin) should be confirmed by demonstrating its usefulness. Hence my questions below for a minor revision:

1.     - The authors write about the lack of differences, including statistically significant ones, when comparing various factors. This raises doubts about how useful and sensitive the tested model is to detect these differences? The article would benefit from adding such information.

2.     - At what level of irritation to human skin, which of the other surfactants would be able to cause changes in the model used?

3.     - Whether the dose/concentration effect was assessed and resulted in any changes in the model?

Comments on the Quality of English Language

Minor editing of English language recommended

Author Response

  1. The authors write about the lack of differences, including statistically significant ones, when comparing various factors. This raises doubts about how useful and sensitive the tested model is to detect these differences? The article would benefit from adding such information. 

Response

The manuscript does discuss the drawback to the use of 3D skin models. One of these drawbacks is the high financial cost of the model. This limited the number of technical repeats we could carry out in this current study which may explain why some the results were only trending and not statistically significant. This was omitted from the original version of the manuscript; however, we have adapted the conclusion to state this and added that this is a factor that requires consideration when choosing an appropriate pre-clinical model to follow on from in vitro 2D cell culture.

  1. At what level of irritation to human skin, which of the other surfactants would be able to cause changes in the model used?

Response

As suggested in the Economic Cooperation and Development (OECD) document (section 24) referenced in our article, up to 5 % aqueous sodium dodecyl sulphate (SDS) is required to affect tissue viability and barrier functions. However, the high cost of the 3D in vitro skin model utilised in the current study limited us to the testing of only two surfactants - one of which was naturally derived biosurfactant (acidic SL) and the other was synthetically derived surfactant (SLES), which was utilised as comparison to the acidic SL. These compounds and the treatment concentration were selected based on previous work we reference in the manuscript on comparing different biosurfactant congeners in 2D in vitro models.

  1. Whether the dose/concentration effect was assessed and resulted in any changes in the model?

Response

As previously stated, the LabSkin 3D in vitro skin model used in this current study was costly from a monetary perspective and as such we were highly limited to the number of experimental conditions, we could subject the model to. The concentration of both SLES and acidic-SL chosen for this study were based upon previous work carried out in our lab using these compounds are varying concentrations on 2D cell models (see several of our publications we reference in the manuscript). As such this study is not a full pre-clinical validation of acid-SL and at no point do we state that it is in the manuscript, only a progression from our previous work and a test bed for trialling these compounds in a 3D in vitro skin model. We do state this in the conclusion section of our manuscript.

Reviewer 2 Report

Comments and Suggestions for Authors

General comment

The manuscript by Adu and coauthors describes the use of crude acidic SLs influence as ingredients for possible topical application in comparison to conventional and more aggressive SLES. The manuscript is well written and easy to read; the data reported show that acidic SLs do not affect the viability of S.epidermis in 3D skin model, the skin morphology or induce any inflammation in comparison to SLES. The authors bring novel and interesting results for the scientific community and conducted a rigorous evaluation and comparison between the two model systems. Nevertheless, some issues are highlighted, as some experimental protocols are referred to but not described in the M&M or in the SI and the description of the results may be difficult for non-experts in the field. 

I would suggest accepting this manuscript after some revisions as follows.

Comments:

Lines 65-69. They could be united as they introduced similar concept in different ways, commenting that 2D systems are not enough to be used as a model for complex 3D in vivo systems.

Lines 107-113. I would recommend to rephrasing them explaining in more focused manner the aim of the study. The authors should stress the use of bacteria to mimic the natural microbiota of the skin, as it does not become evident until the mid-section of the manuscript.

Line 118. The author should specify the buffer they used. 

Lines 154-155. The instrumental protocol for ESI MS is missing and should be added or a reference should be added at least.

Line 168. CFU is introduced already in line 147, so the acronym should be introduced there.

Line 282. I am a bit skeptical in reading that 96.41 % (lack of SD?) of sample with different acidic SLs is considered pure. I understand that the authors want to prove that a crude sample can be applied as well as a purified one, but the term pure is misleading by a chemical point of view (as also shown by Figure 2 and table 1). I would suggest to rephrasing it underlining the ratio between the acidic and lattonic form but not using the word pure which can be controversial.

Table 1. The text on the right column is cut.

Figure 3. Instead of using the measurement units only in the y axis label, the author should state what they are plotting vs the conditions of the samples.

Lines 325. I would recommend using the term “untreated” instead of “non-surfactant”.

Lines 326-329 and Figure 4. Why do the author state that there is not visible change in morphology in the layers of the skin after treating them in the different conditions? It seems that the (apparent) roughness changes in c and slightly in d in comparison to a and b, or is it considered a normal morphology anyway? Especially for the non-experts in the field, the purple dots (fibroblast?) should be described in the micrographs. 

Lines 339-340. All the values presented should have the correct number of significant figures for the data points and the SDs.

Lines 346-349. As in my previous comment about the change in morphology, Figure 5 shows a change in the curvature of the skin upon exposure to SLs and SLES (stronger in this case) in comparison to the control, should this be explained or is this considered normal depending on the section of the skin examined?

Lines 405-407. I would recommend rephrasing it stating directly that the difference in terms of CXCL8 concentration is not significant among the samples S.epidermis colonized and non-treated. 

Author Response

Comments:

 Lines 65-69. They could be united as they introduced similar concept in different ways, commenting that 2D systems are not enough to be used as a model for complex 3D in vivo systems.

Response

We agree with the reviewer and have removed the paragraph insert between lines 65-69 and removed some text around line 69 to consolidate the explanation of drawbacks to 2D in vitro models. We have also altered some of the text where we discuss the use of in vivo animal models to generate pre-clinical data.

Comment

Lines 107-113. I would recommend to rephrasing them explaining in more focused manner the aim of the study. The authors should stress the use of bacteria to mimic the natural microbiota of the skin, as it does not become evident until the mid-section of the manuscript.

Response

We have re-written aims paragraph of the manuscript to clearly state the overriding aim of the study which is to progress our previous research in the area on 2D cell culture models. In doing so we have stated why S. epidermidis was selected for skin mode colonisation (due to being a skin commensal organisms). We have additionally altered the text to provide the reader a clearer outline of the studie’s general methodology.

Comment

Line 118. The author should specify the buffer they used.

Response

Done – sterile Phosphate Buffered Saline

Comment

Lines 154-155. The instrumental protocol for ESI MS is missing and should be added or a reference should be added at least.

Response

In section 2.1 “Chemical characterisation of Acidic SL” we state that ASL used in the study was purchased from an outside supplier and then chemically characterised in house using HPLC–MS/ESI. We provide a reference for this method from our previous work in the field “Adu, S.A.; Twigg, M.S.; Naughton, P.J.; Marchant, R.; Banat, I.M. Biosurfactants as Anticancer Agents: Glycolipids Affect Skin Cells in a Differential Manner Dependent on Chemical Structure. Pharm. 2022, Vol. 14, Page 360 2022, 14, 360, doi:10.3390/PHARMACEUTICS14020360.” Referce 33 in the orginal manuscript. We therfore respectivly feel that additional explination is not required in section 2.4.

Comment

Line 168. CFU is introduced already in line 147, so the acronym should be introduced there.

Response

Corrected

Comment

Line 282. I am a bit skeptical in reading that 96.41 % (lack of SD?) of sample with different acidic SLs is considered pure. I understand that the authors want to prove that a crude sample can be applied as well as a purified one, but the term pure is misleading by a chemical point of view (as also shown by Figure 2 and table 1). I would suggest to rephrasing it underlining the ratio between the acidic and lattonic form but not using the word pure which can be controversial.

Response

We understand the point that the reviewer is making and have tried to replace the term pure with the more precise percentage relative abundance of sophorolipid chemical sub-species in the revised manuscript. In general, our point in determining and highlighting the high percentage of Acidic SL in the prep used in this study is to show the importance of obtaining  samples that possesses a high levels in percentage abundance of the sophorolipid sub-species in question when investigating bioactivity. This unfortunately is not something that has been considered in many previous studies within the field of biosurfactant bioactivity.

Comment

Table 1. The text on the right column is cut.

Response

This seems to have been an issue that has arose when converting the manuscript word document to pdf, therefore we have adapted Table 1 to avoid this.

Comment

Figure 3. Instead of using the measurement units only in the y axis label, the author should state what they are plotting vs the conditions of the samples.

Response

We have adapted figure 3 in line with the reviewer’s suggestion.

Comment

Lines 325. I would recommend using the term “untreated” instead of “non-surfactant”.

Response

These terms have been changed.

Comment

Lines 326-329 and Figure 4. Why do the author state that there is not visible change in morphology in the layers of the skin after treating them in the different conditions? It seems that the (apparent) roughness changes in c and slightly in d in comparison to a and b, or is it considered a normal morphology anyway? Especially for the non-experts in the field, the purple dots (fibroblast?) should be described in the micrographs.

Response

The authors would like to clarify that there was no observable difference in the morphology of the untreated/ treated tissues. The apparent roughness/ changes in the micrographs were as a result of machine-tissue handling and processing, which were seen in most samples. Normally, morphological changes of interest should be ulcerations, tissue erosion, and necrosis, all of which were absent post tissue treatment under all conditions.

The composition of the dermal fibroblast has been briefly described in the micrograph as suggested (figure 4).

Comment

Lines 339-340. All the values presented should have the correct number of significant figures for the data points and the SDs.

Response

We were unsure as to what the reviewer wants in this comment? However, we have adapted the manuscript to add um to the SDs and added the p values for the statistical analysis of the data. We hope this satisfies the reviewer’s comment.

Comment

Lines 346-349. As in my previous comment about the change in morphology, Figure 5 shows a change in the curvature of the skin upon exposure to SLs and SLES (stronger in this case) in comparison to the control, should this be explained or is this considered normal depending on the section of the skin examined?

Response

Please refer to previously addressed comment on this.

We would like to add that samples for immunohistochemistry staining undergo further processes prior to coating with primary and secondary antibodies (please check sub-section 2.8 of the manuscript for detailed methodology) and this may have caused the curvature of concern. Nonetheless, the focus of tissue analysis at this stage of study was more on stain intensity of biological marker of interest (filaggrin) than tissue morphology.

Comment

Lines 405-407. I would recommend rephrasing it stating directly that the difference in terms of CXCL8 concentration is not significant among the samples S.epidermis colonized and non-treated.

Response

This section of the manuscript is concerned with gene expression as measured by RT-qPCR and as such the measurement being compared is not the concentration of CXCL8 (the gene encoding the IL-8 cytokine) but its fold change in expression. We have attempted to modify the text of this section the manuscript relating to gene expression changes to make this clearer.